# Viscoelastic Parameters of Asphalt Mixtures Identified in Static and Dynamic Tests

**DOI:** 10.3390/ma12132084

**Published:** 2019-06-28

**Authors:** Piotr Mackiewicz, Antoni Szydło

**Affiliations:** Faculty of Civil Engineering, Wrocław University of Science and Technology, 50-370 Wrocław, Poland

**Keywords:** viscoelastic parameters, creep test, fatigue tests, asphalt mixtures, Burgers model, four point bending beam

## Abstract

We present two methods used in the identification of viscoelastic parameters of asphalt mixtures used in pavements. The static creep test and the dynamic test, with a frequency of 10 Hz, were carried out based on the four-point bending beam (4BP). In the method identifying viscoelastic parameters for the Brugers’ model, we included the course of a creeping curve (for static creep) and fatigue hysteresis (for dynamic test). It was shown that these parameters depend significantly on the load time, method used, and temperature and asphalt content. A similar variation of parameters depending on temperature was found for the two tests, but different absolute values were obtained. Additionally, the share of viscous deformations in relation to total deformations is presented, on the basis of back calculations and finite element methods. We obtained a significant contribution of viscous deformations (about 93% for the static test and 25% for the dynamic test) for the temperature 25 °C. The received rheological parameters from both methods appeared to be sensitive to a change in asphalt content, which means that these methods can be used to design an optimal asphalt mixture composition—e.g., due to the permanent deformation of pavement. We also found that the parameters should be determined using the creep curve for the static analyses with persistent load, whereas in the case of the dynamic studies, the hysteresis is more appropriate. The 4BP static creep and dynamic tests are sufficient methods for determining the rheological parameters for materials designed for flexible pavements. In the 4BP dynamic test, we determined relationships between damping and viscosity coefficients, showing material variability depending on the test temperature.

## 1. Introduction

The selection of correct material parameters is very important, both in engineering practice and scientific study. The determination of reliable material properties is also essential in further structural analyses. The appropriate material parameters and the model enable the use of efficient numerical methods, and determine the state of stresses and deformations in the construction model. It is especially important for asphalt mixtures used as the main material in vulnerable road pavements. Such mixtures are thermo-rheological, changing their properties under thermal conditions and load time. In various conditions, both under static and dynamic loads, they reveal their rheological characteristics. These properties are much more important in the description of the material in higher temperatures than in lower ones, in which linear–elastic models are sufficient to model the material parameters. The asphalt layers in the road pavements show both elastic and viscous features. The elastic properties dominate at the lower temperatures, and are responsible for irreversible deformations of the asphalt pavement, whereas the viscous features are typical of the higher temperatures. Therefore, proper identification of the rheological parameters of asphalt mixtures based on the results of laboratory tests is not easy. The typical static tests in which these parameters are defined include the static creep, testing under the constant load when the cylindrical specimens are compressed and the beams are bent [1]. Dynamic tests are analogous, testing with compressed cylindrical specimens [2] and bend fatigue beams. 

Different rheological parameters can be obtained in various mounting schemes and load conditions, characterized by duration and frequency. Therefore, the choice of proper research method is important. This method is used to determine these parameters and the models describing the behaviour of the structure. In the case of road pavements, both static (parking lots, crossroads, etc.) and repetitive loads with short-term impact are analysed. As mentioned earlier, the asphalt mixtures become viscous over time in high temperatures for a long-term static load, whereas the accumulation of permanent deformations resulting in permanent deformation (i.e., ruts) occurs under dynamic loading. However, changing the viscoelastic dissipative energy is also important in fatigue tests. This change significantly affects the fatigue destruction of the material. This publication analyses the behaviour of the asphalt mixture under the static and dynamic loading for a four-point bending beam (4BP).

## 2. Identification of Rheological Parameters

Many rheological models are used in the common road practice. As already mentioned, the asphalt mixtures expose their rheological properties at high temperatures. The viscoelastic models are used to describe these properties. The viscoelasticity theory is increasingly being used in the analysis of asphalt pavements, due to its good description of flow and deformation of road materials.

According to Reiner and Ward [3], the first papers about rheology come from the thirties of the previous century. However, this discipline has intensively developed since the 1950s [4], and deals with materials and constructions of buildings as well as road pavements. In the fundamental work edited by Reiner and Ward [3], there is a chapter devoted to the rheology of materials and asphalt pavements written by Van der Poela [5]. Regarding asphalt mixtures, one of the first works by Monismith et al. [6] deserves special attention. The authors found that asphalt pavement mixtures also exhibit linear viscoelastic properties at very low deformations. By studying the creep of the asphalt pavement mixtures, Vakili [7] draws the same conclusions. Goodrich [8] studied asphalt mixtures with mineral fillers, as well as the large aggregate under oscillations with small amplitudes, and found again that these materials show linear viscoelastic features at very small deformations.

In theoretical considerations, Kisiel and Lysik [9], Nowacki [10], and Jakowluk [11] contributed significantly to development the rheology in construction. The use of rheological models in the description of asphalt mixtures can be found, among others, in the works [12,13,14,15]. The identified rheological parameters were also studied under different static [16,17,18,19] and dynamic load conditions [6,13,20]. However, no comprehensive comparisons have been made to the four-point study of static and dynamic conditions, although the 4BP is commonly used. There are also no comparisons to other various laboratory studies.

Currently, there are many analytic methods [21,22,23] and numerical models, including micromechanical models [24] and anisotropic models [25,26], in which the material parameters of asphalt mixtures are used in the assessment of the behaviour of flexible pavement.

Currently, due to the high availability of software for numerical calculations, no attention is paid to the selection of the appropriate research method, the application of the appropriate model, or the use of valid parameters in the models of surfaces. Moreover, the entire creep curve is not included with the load curve in the determination of parameters. Both simple and complex rheological models were analysed. For example, the complex constitutive models, with and without damage, can be found in [27,28,29,30,31]. Other studies have addressed advanced pavement structural models with and without dynamic effects [32,33,34].

It has been found that the Burgers model, among many other viscoelastic models, reliably describes asphalt concrete behaviour [12,17,35,36]. The model diagram along with its parameters is shown in Figure 1.

The study of static creeping was performed under the 4BP bending conditions. The creep curve in the Burgers model has its graphic interpretation, shown in Figure 2. Parameters can be determined by immediate deformations, maximum deformations (elastic moduli), and the rate of deformations (viscosity coefficients). However, such interpretation is not very accurate, because large errors can occur when the immediate deformations are registered during elastic recurrence. Therefore, we proposed to determine these parameters using numerical methods, taking into account the overall creep curve at loading and the curve at unloading.

A conjugated gradient method was used to approximate the laboratory creep curve using the theoretical curve. This is an effective method for solving optimization problems. The minimum of function was determined from each point in a given search direction. The rheological parameters of the Burgers model were the sought variables: *E*_1_, *E*_2_, *η*_1_, *η*_2_ (see Figure 1). The target function is described by Equation (1):(1)$$\Delta p=\sqrt{\frac{\sum_{i=1}^{i=l}{\left(\varepsilon_{ti}-\varepsilon_{li}\right)}^{2}}{l}}100\%$$
where *ε_li_* is the deformation measured on the sample, *ε_ti_* is the theoretical deformation calculated for the model, and *l* is the number of measured points.

The theoretical deformations were determined from the equations of the Burgers model, starting from the differential constitutive relationship between stress σ and deformation ε:(2)$$\sigma +a\dot{\sigma }+b\overset{¨}{\sigma }=c\dot{\varepsilon }+d\overset{¨}{\varepsilon }$$
(3)$$\sigma +\left(\frac{\eta_{1}}{E_{1}}+\frac{\eta_{1}}{E_{2}}+\frac{\eta_{2}}{E_{2}}\right)\cdot \dot{\sigma }+\frac{\eta_{1}\eta_{2}}{E_{1}E_{2}}\overset{¨}{\sigma }=\eta_{1}\dot{\varepsilon }+\frac{\eta_{1}\eta_{2}}{E_{2}}\overset{¨}{\varepsilon }$$

After the solution of these equations, the relationship between the deformation ε(*t*) and the time *t* was obtained Equations (4) and (5):(4)$$\mathrm{for}\;\mathrm{the}\;\mathrm{load}\;t\;<\;t_{0}\;\varepsilon (t)=\sigma_{0}\left[\frac{1}{E_{1}}+\frac{t}{\eta_{1}}+\frac{1}{E_{2}}\left(1-e^{\frac{-tE_{2}}{\eta_{2}}}\right)\right],$$
(5)$$\mathrm{or}\;\mathrm{the}\;\mathrm{unload}\;t\;>\;t_{0}\;\varepsilon (t)=\sigma_{0}\left[\frac{t_{0}}{\eta_{1}}-\frac{1}{E_{2}}e^{\frac{-tE_{2}}{\eta_{2}}}\left(1-e^{\frac{t_{0}E_{2}}{\eta_{2}}}\right)\right]$$

The identification of rheological parameters can effectively contribute to the optimization of mixture composition also under fatigue conditions, when there exists energy dissipation due to microcracks. The procedure for determining the rheological parameters under repetitive stress conditions was performed. In this test, the parameters were determined at the 10 Hz load frequency. This frequency was applied according to the European Standard EN 12697-24:2012 [37], in order to evaluate the fatigue characteristics of asphalt mixtures. The identification of the parameters was performed by the selection of parameters in the Burgers model for hysteresis, describing the relationship between stress σ and deformation ε (Figure 3), and using the conjugate gradient method to minimize the function in Equation (1).

The controlled amplitude of displacement and the time of its delay in relation to the acting force were recorded directly on the basis of the fatigue strength test, using the variable force. Based on the basic dependencies for the 4BP beam (Equations (6)–(8)), it is possible to determine the required stress value σ, deformation ε, and phase angle *ϕ* in any cycle and at any time of the load, as follows:(6)$$\sigma =\frac{3Pa}{bh^{2}}$$
(7)$$\varepsilon =\frac{12\Delta h}{3L^{2}-4a^{2}}$$
(8)φ=360fs
where *P* is the force (N); *b* and *h* are the beam width and height, respectively (m); *a* is the distance between the support and the force (m), and *a* = *L*/2; *Δ* is the displacement (m); *L* is the spacing of the supports (m); *f* is the frequency (Hz), and *f* = *ω*/2π; and *s* is the delay time between the force *P* and the displacement *Δ* (s).

According to Figure 3, it is possible to determine the complex stiffness modulus *E** and phase angle *φ* between deformation and stress:(9)$$E*=\frac{\sigma_{0}\sin\left(\omega t\right)}{\varepsilon_{0}\sin\left(\omega t-\varphi \right)}$$
(10)$$tg\varphi =\frac{E_{2}}{E_{1}}$$

Additional conditions for the agreement between the phase angle and the composite modulus found in the test and model were introduced in the search criteria for the most accurate matching of the laboratory results, with the hysteresis determined by the Burgers model. The dependence of the phase angle and the complex modulus on the parameters in the Burgers model are described by Equations (11) and (12):(11)$$E*=\omega {\left[\frac{c^{2}+{\left(d\omega \right)}^{2}}{{\left(b\omega^{2}-1\right)}^{2}+{\left(a\omega \right)}^{2}}\right]}^{1/2}$$
(12)$$tg\varphi =\frac{ad\omega^{2}-\left(b\omega^{2}-1\right)c}{\left(b\omega^{2}-1\right)d\omega +ac\omega }$$

Using Equation (2), for the cyclic symmetrical sinusoidal deformation *ε* = *ε*_0_ sin(*ωt* − *ϕ*), we obtain the relationship
(13)$$\sigma +a\dot{\sigma }+b\overset{¨}{\sigma }=-\varepsilon_{0}\omega \left[d\omega \sin\left(\omega t\right)-c\cos\left(\omega t\right)\right]$$

The variables *a*, *b*, *c*, and *d* present in the constitutive Equation (13) are described by Equations (14)–(17):(14)$$a=\frac{\eta_{1}}{E_{1}}+\frac{\eta_{1}}{E_{2}}+\frac{\eta_{2}}{E_{2}}$$
(15)$$b=\frac{\eta_{1}\eta_{2}}{E_{1}E_{2}}$$
(16)$$c=\eta_{1}$$
(17)$$d=\frac{\eta_{1}\eta_{2}}{E_{2}}$$

## 3. Materials and Methods

The static creeping test with 4BP bending was performed on the NAT (Nottingham Asphalt Tester, University of Nottingham, Nottingham, UK) device, which enables the efficient testing of many asphalt mixtures under various mounting and loading patterns. This device is characterized by the good reproducibility of results. The technical conditions for the study were adopted according to manual (Cooper Research Technology [38]). The following conditions were applied: a constant load with 0.30 MPa (15% of the bending strength at 25 °C), a load time of 1800 s, and an unload time of 510 s. In order to determine rheological parameters, we applied four temperatures: −5 °C, 0 °C, 10 °C, and 25 °C. The dimensions of the samples were as follows: the width was 60 mm, the height was 50 mm, and the length was 384 mm. In Figure 4, a schematic diagram of the static 4BP creep test is shown. 

The 4BP dynamic test consists of the cyclic bending of the beam supported in four points, as shown in Figure 5 (accordance with EN 12697-24:2012 [37]). Due to common research practice, the study was conducted under the sinusoidal kinematic constraints, with the controlled deformation. The amplitude of deformation was 100 × 10^−6^. This method allows us to compare the received results to the known fatigue criteria in the design practice [35,39]. The dimensions of the bending beams and the temperature conditions were assumed as for the static testing. The basic parameter that was determined during the test was the fatigue hysteresis, which depends on the number of load cycles. The tests with the fixed peak-to-peak strain allowed us to record the change in stresses in relation to the load cycles.

The research was carried out with Cooper Research Technology Ltd. Beam-Flex, on typical asphalt mixture commonly used in building road pavements, which was laid on the binding surfaces AC16W with asphalt 35/50. Mixtures with different asphalt content were analyzed—i.e., 4.0%, 4.5%, and 5.3%. The formulas of the mixtures were previously designed in accordance with the current technical requirements.

## 4. Results of the Parameters’ Identification

Based on the presented procedure for the identification of the viscoelastic parameters and the tests performed under various temperature conditions and asphalt content, we derived the parameters of the Burgers model for the creep study at static (Table 1) and dynamic (Table 2) loading.

For a mixture with asphalt content 4.5%, the results of study and the approximation of curves using the Burgers creep model in the static test for various temperatures are shown in Figure 6, whereas the results of the dynamic test, as well as the approximation of curves *σ*–*ε* using the Burgers model, are presented in Figure 7, Figure 8, Figure 9 and Figure 10.

It may be noticed that the obtained parameters differ. Moduli of instant elasticity *E*_1_ obtained in the dynamic 4BP test are about 5 to 13 times larger than those received in the static test. Similarly, moduli of delayed elasticity *E*_2_ are about 5 to 7 times greater in the dynamic than in the static tests. On the other hand, viscosity coefficients *η*_1_ and *η*_2_ are about two and three times smaller in the dynamic test than in the static test, respectively. This results from the very short time of variable loading, and consequently, of the short time of the material deformation response. A comprehensive comparison of changes in parameter values for different temperatures and asphalt content is shown in Figure 11 (a static test) and Figure 12 (a dynamic test).

For the temperature and asphalt content range analysed, changes in the parameters were observed. For lower temperatures, a smaller change in parameters is observed depending on the asphalt content. For temperature 25 °C, the largest parameter values (except for *η*_2_) were obtained for the optimal asphalt content 4.5%. For other temperatures, there are less pronounced extremes associated with the composition of the mixture.

Moreover, it is worth noting as the load time increased in the creep static test, smaller values of the elastic parameters *E*_1_ and *E*_2_ were obtained, whereas the viscosity parameters *η*_1_ and *η*_2_ were higher. In the dynamic test, the response of material influenced by the short-variable load was more elastic. However, the rheological features were visible over the entire range of analysed temperatures. It is worth noting that the values of parameters, mainly related to the viscosities *η*_1_ and *η*_2_, were correlated with the phase angle *ϕ*. Its value increases with increasing temperature. The angle change in the low temperature range is practically linear (Figure 13). At higher temperatures and higher angles, the material will have a larger contribution of viscous rather than elastic characters, while the lower the angle, the more elastic the material. Moreover, at higher temperatures, there is a greater variation in the angle value depending on the asphalt content in the mixture. For 5.3% asphalt content, the highest values of the phase angle *ϕ* were obtained.

## 5. Numerical Verification of Rheological Parameters

For a selected mixture with the optimum asphalt content of 4.5%, the numerical verification of rheological parameters was performed using the finite element method. Three-dimensional static and dynamic models were developed. They included the appropriate sample geometry and load conditions that are consistent with the previously described test procedure (Figure 14). Previously, we analysed the division of the model into finite elements. The dimensions of the model were in agreement with those of the laboratory tested sample. To build the model, we used 410,000 eight-node volume elements. In the middle part of the beam, the density of the element grid was greater. Such discretization allowed for the convergence of results for displacements and deformations. The calculation of the model was carried out in the SOLIDWORKS-COSMOS/M software, ver. 2010, Structural Research and Analysis Corporation, Santa Monica, CA, USA.

Rheological parameters were appropriately applied for the static and dynamic testing, according to the date presented in Table 1; Table 2. In the dynamic study, we assumed the density to be 2400 kg/m^3^, with appropriate damping parameters associated with the dynamic analysis included in the static testing. The dynamic problem of discretization in Finite Element Method (FEM) is described by the classical equation [40]
(18)$$\left[M\right]\left\{\ddot{u}(t)\right\}+\left[C\right]\left\{\dot{u}(t)\right\}+\left[K\right]\left\{u(t)\right\}=\left\{F(t)\right\}$$
where: [*M*] is the matrix mass, [*C*] is the damping matrix, [*K*] is the stiffness matrix, {*F*(*t*)} is the load vector variable in time *t*, $\left\{\ddot{u}(t)\right\}$ is the acceleration vector in the time *t*, $\left\{\dot{u}(t)\right\}$ is the velocity vector in time *t*, and {u(t)} is the displacement vector in time *t*.

Selecting appropriate damping for the material is an important issue in the dynamic model. This is a complex phenomenon that involves the dissipation of energy through a variety of mechanisms, such as internal friction, cyclic thermal effects, microscopic material deformation, and micro- and macrocracks. The damping process consists of damping material, structural damping, and viscous damping associates with energy dissipation. To realistically simulate the material behavior under a short-term load, the damping factor is included as an important parameter. This is difficult to model, but the existing damping models are available in numerical calculation programs. The damping models can depend on the frequency or viscosity. The Rayleigh’s damping model is quite often used in the structural dynamic analysis. To include damping effects, the damping coefficients *α* and *β* should be calculated. They are present in Rayleigh’s damping matrix [40]:(19)[C]=α[M]+β[K]

The damping coefficients are related to the angular frequency, in the form of Rayleigh’s damping coefficient:(20)$$\xi =\frac{\alpha }{2\omega }+\frac{\beta \omega }{2}$$

At present, no effective experimental methods have been developed to identify damping parameters for asphalt mixtures. There is no simple correlation between the damping and static or dynamic deflections. In practice, it is possible to use FEM, and it is enough to associate only the damping with the rigidity of the system [41]. The damping was also applied for this beam model, but with only the stiffness obtaining satisfactory results. The damping parameter *β* was identified in the model using iterative back-calculations maintaining the agreement between the deformations from the laboratory studies. The following damping parameters *β* were obtained: 0.002 1/s for −5 °C, 0.0025 1/s for 0 °C, 0.004 1/s for 10 °C, and 0.005 1/s for 25 °C. Correlation between damping parameters and viscosity coefficients was determined (Figure 15). We have described the relationships between viscosity parameters and the damping coefficient using linear regression functions. High correlation coefficients (close to 1) were obtained for this function. As the value of viscosity parameters increases, the value of damping coefficients decreases.

Figure 16 shows the results of deformation for the static creep tests for different temperatures, and Figure 17 presents the results for the fatigue dynamic test.

The numerical analysis allows us to indicate the proportion of elastic and viscous deformations depending on the test temperature. It has been found that with increasing temperature, the proportion of viscous deformations also increases, and in a dynamic temperature test is 6% at −5 °C, 8% at 0 °C, 11% at 10 °C, and 25% at 25 °C. The results for the creeping test are 19% at −5 °C, 37% at 0 °C, 71% at 10 °C, and 93% at 25 °C. The differentiation of the contribution of weak deformations results from different time intervals of the load in both studies. It is worth noting, however, that in the case of the static creep testing, the increase in temperature results in a significant nonlinear growth in the value of viscous deformations.

## 6. Conclusions

The different values of viscoelastic parameters in a Burgers model were determined. The variability of the parameters in temperature was obtained for the static and dynamic tests.
These parameters depended significantly on the duration of the load. Therefore, appropriate parameters should be chosen depending on the load time when the behaviour of asphalt mixtures in the pavement is modelled.For the static long-term load tests, the parameters should be derived from creep curves, and for dynamic tests, they should be determined from the hysteresis.It was found that the use of the Burgers viscoelastic model is justified for dynamic loads with the frequency of 10 Hz. For higher frequencies and at lower temperatures, the determination of the parameters may be of lesser importance, because the material has parameters similar to the elastic model, due to its low phase angle.The creep test using static and dynamic 4BP loading is an effective method for determining rheological parameters under the assumed load time, the number of cycles, and temperatures. The linear viscoelastic Burgers model is helpful in this regard, because interprets the thermoplastic features of the road pavement material, such as the asphalt mixtures, well.The numerical analysis using the finite element method allows us to identify the contribution of viscous deformations relative to the total, and show the significant variation of these deformations for two tests, according to the temperature.The rheological parameters also depend on the composition of the bituminous mixture. For the optimal asphalt content (4.5%), the highest values of rheological parameters were obtained, demonstrating the best mechanical features and resistance to permanent deformations. For the increased asphalt content, viscosity coefficients clearly decrease, which corresponds to the increase in the value of phase angle *ϕ* and material damping values.The obtained rheological parameters from both methods proved to be sensitive to a change in asphalt content, which means that the methods can be used to design the optimal asphalt mixtures composition—e.g., due to permanent deformation of road surfaces.

In further publications, the calculations using the finite element method for both tests, taking into account the Burgers model, will be verified. In addition, the Burgers parameters will be analysed in the dynamic fatigue test. These parameters change due to the dissipation processes and structural variation in the material.

## Figures and Tables

**Figure 1 materials-12-02084-f001:**
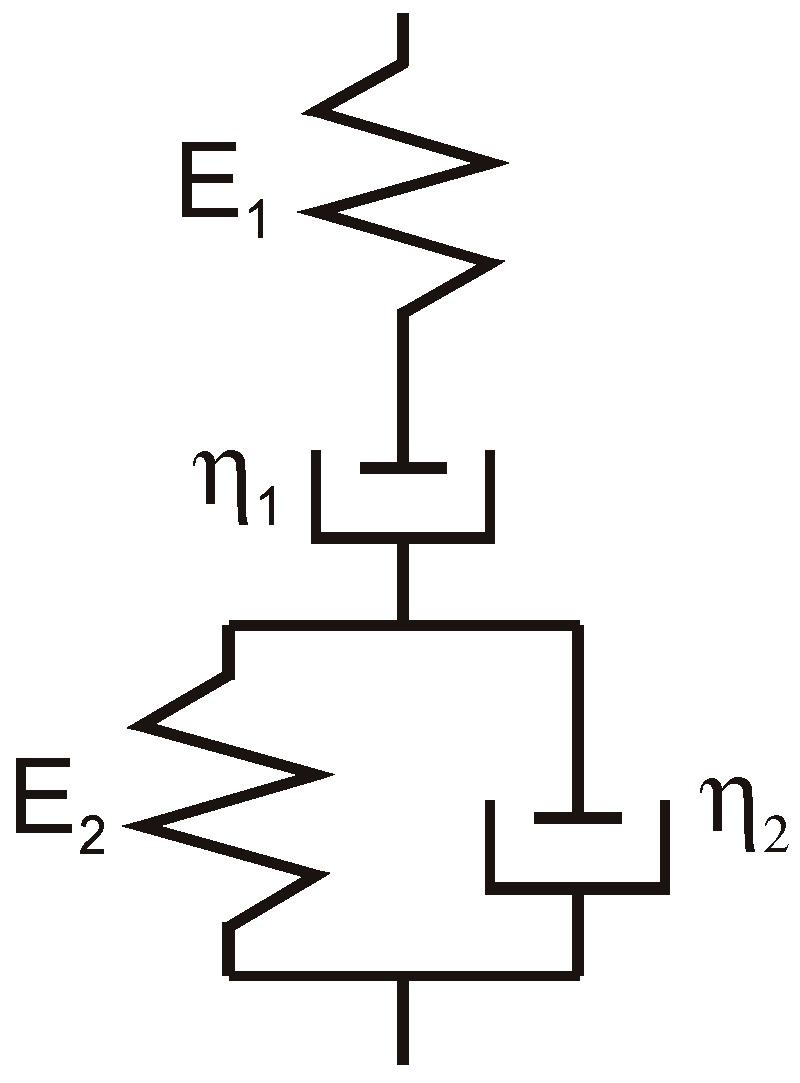
Burgers rheological model. *E*_1_: immediate elastic modulus in the Burgers model (Pa); *E*_2_: delayed elastic modulus in the Burgers model (Pa); *η*_1_: viscosity coefficient in the Burgers model (Pa·s), *η*_2_: viscosity coefficient of elastic delay in the Burgers model (Pa·s).

**Figure 2 materials-12-02084-f002:**
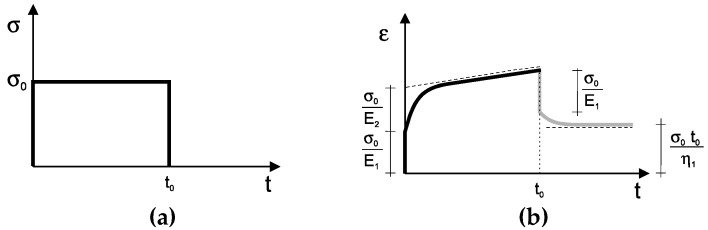
The creep curve in the Burgers model. The graphical finding of the parameters is presented. (**a**)—stress vs. time curve, (**b**)—strain vs. time curve.

**Figure 3 materials-12-02084-f003:**
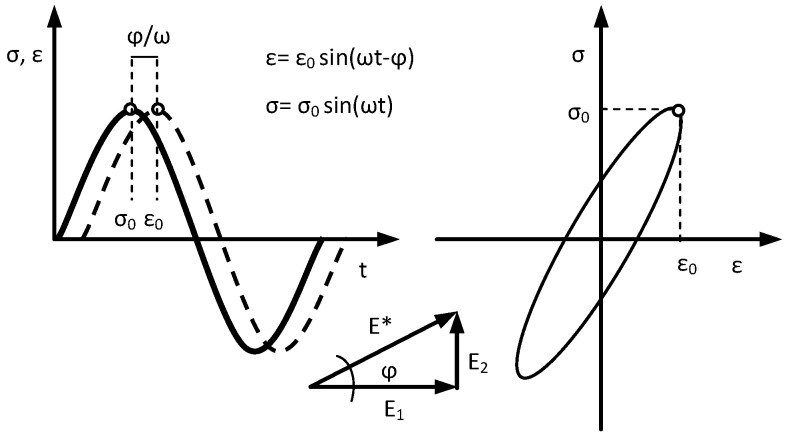
Fatigue hysteresis used for the identification of parameters. *ϕ*: phase angle (°); *ω*: angular frequency = 2π*f* (1/s); *t*: time (s); *σ*_0_: the amplitude of stress (MPa); *ε*_0_: the amplitude of deformation (-); *E**: composite modulus (MPa); *E*_1_: the real element of the composite modulus (MPa); *E*_2_: the imaginary element of composite modulus (MPa).

**Figure 4 materials-12-02084-f004:**
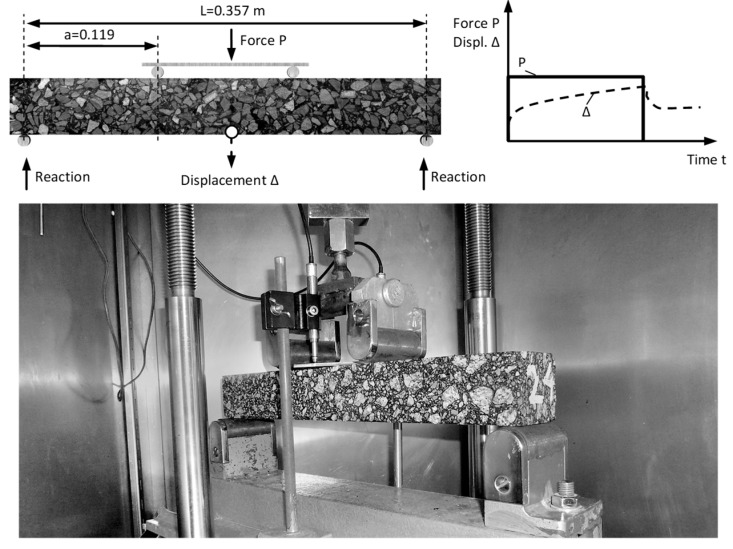
The scheme of the static four-point bending beam (4BP) creep test.

**Figure 5 materials-12-02084-f005:**
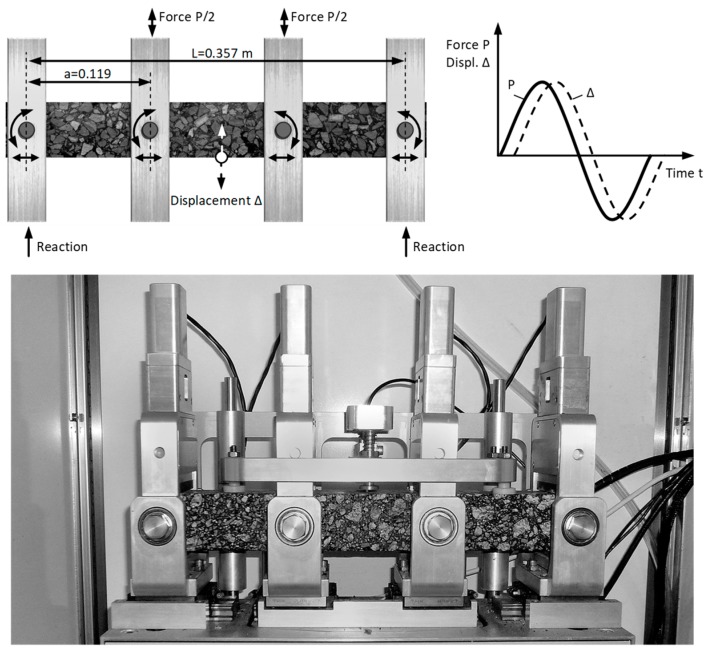
The scheme of the dynamic 4BP test.

**Figure 6 materials-12-02084-f006:**
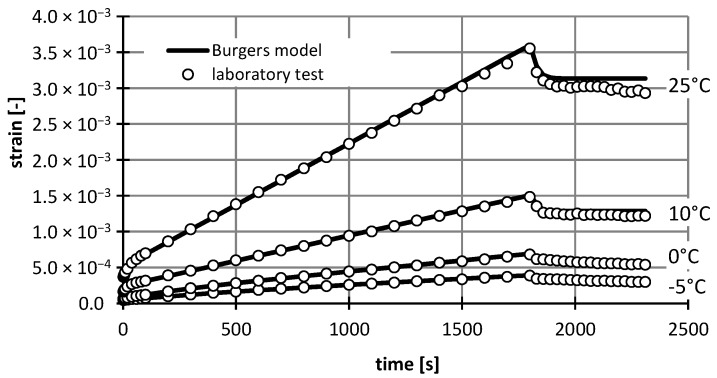
The results of laboratory study and the approximation of creep curves using the Burgers model in the static test (asphalt mixtures with an asphalt content of 4.5%).

**Figure 7 materials-12-02084-f007:**
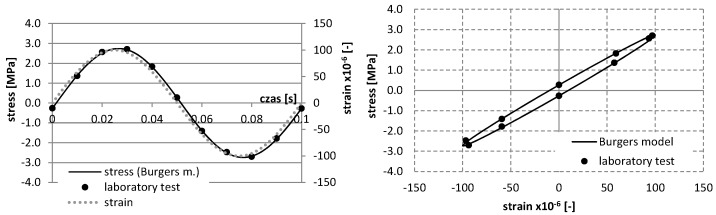
The results of laboratory study and the approximation of curves σ–ε using the Burgers model in the dynamic test at the temperature −5 °C (asphalt mixtures with an asphalt content of 4.5%).

**Figure 8 materials-12-02084-f008:**
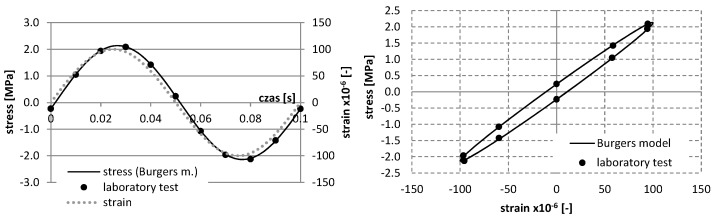
The results of laboratory study and the approximation of curves σ–ε using the Burgers model in the dynamic test at the temperature 0 °C (asphalt mixtures with an asphalt content of 4.5%).

**Figure 9 materials-12-02084-f009:**
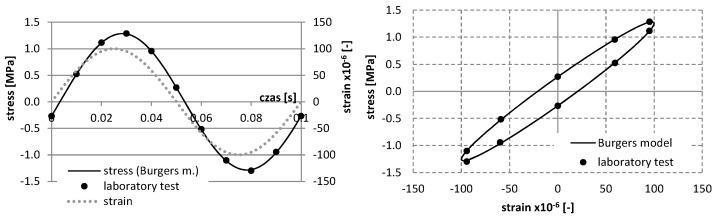
The results of laboratory study and the approximation of curves σ–ε using the Burgers model in the dynamic test at the temperature 10 °C (asphalt mixtures with an asphalt content of 4.5%).

**Figure 10 materials-12-02084-f010:**
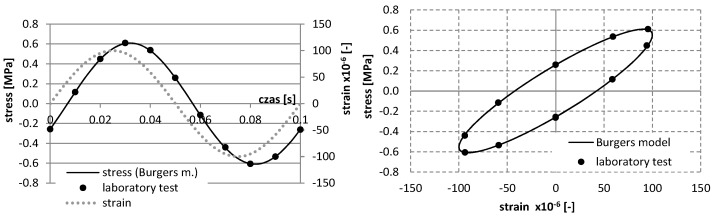
The results of laboratory study and the approximation of curves σ–ε using the Burgers model in the dynamic test at the temperature 25 °C (asphalt mixtures with an asphalt content of 4.5%).

**Figure 11 materials-12-02084-f011:**
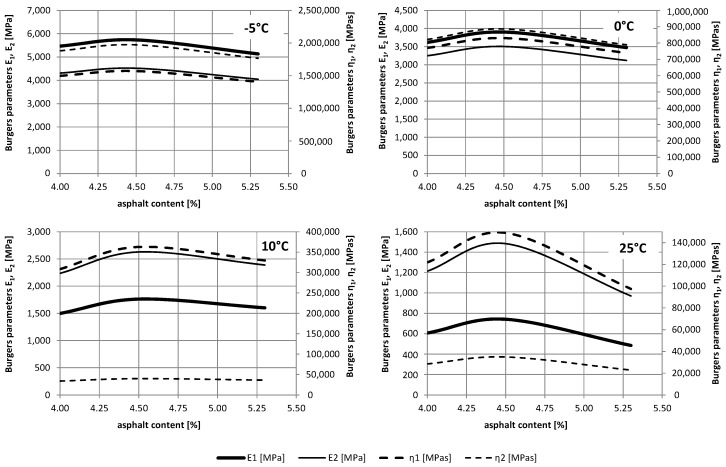
The relationship of Burgers parameters on the temperature in the static test.

**Figure 12 materials-12-02084-f012:**
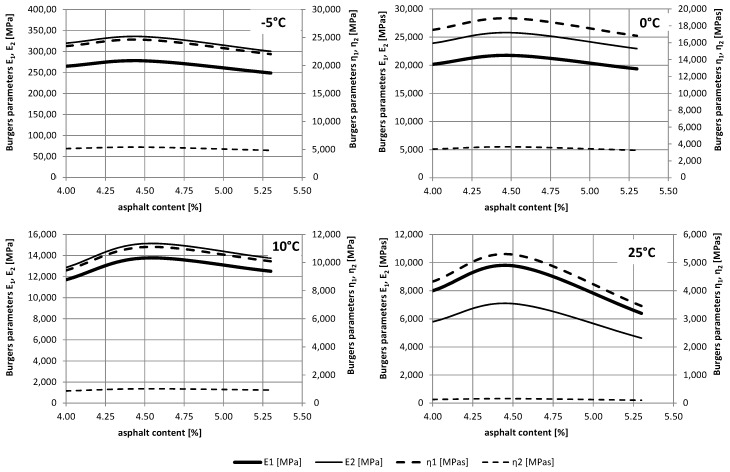
The relationship of Burgers parameters on the temperature in the dynamic test.

**Figure 13 materials-12-02084-f013:**
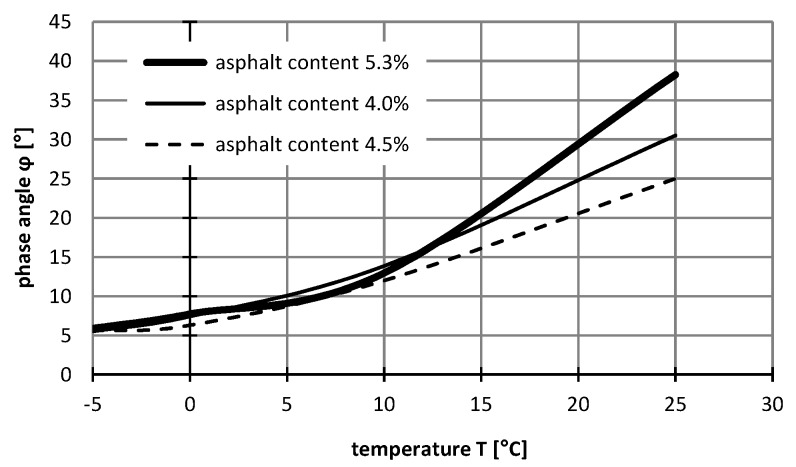
The relationship between the phase angle and temperature.

**Figure 14 materials-12-02084-f014:**
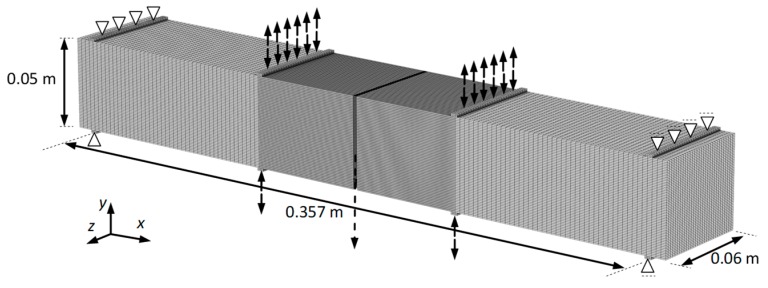
The relationship between the phase angle and temperature.

**Figure 15 materials-12-02084-f015:**
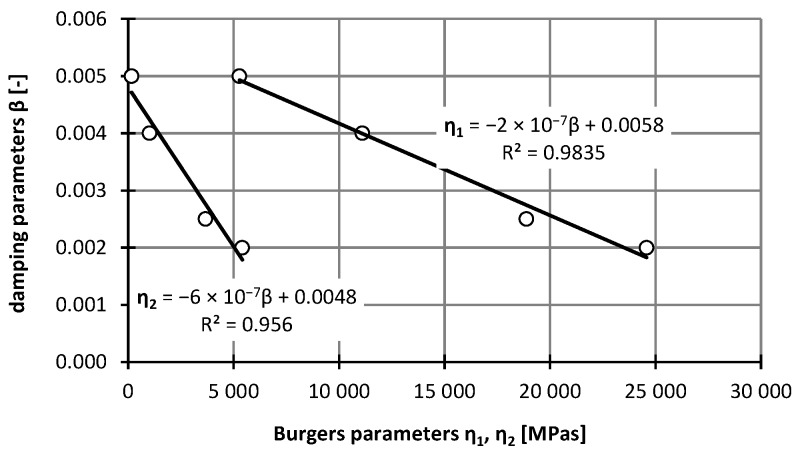
The relationship between the phase angle and the damping parameters (asphalt mixtures with an asphalt content of 4.5%).

**Figure 16 materials-12-02084-f016:**
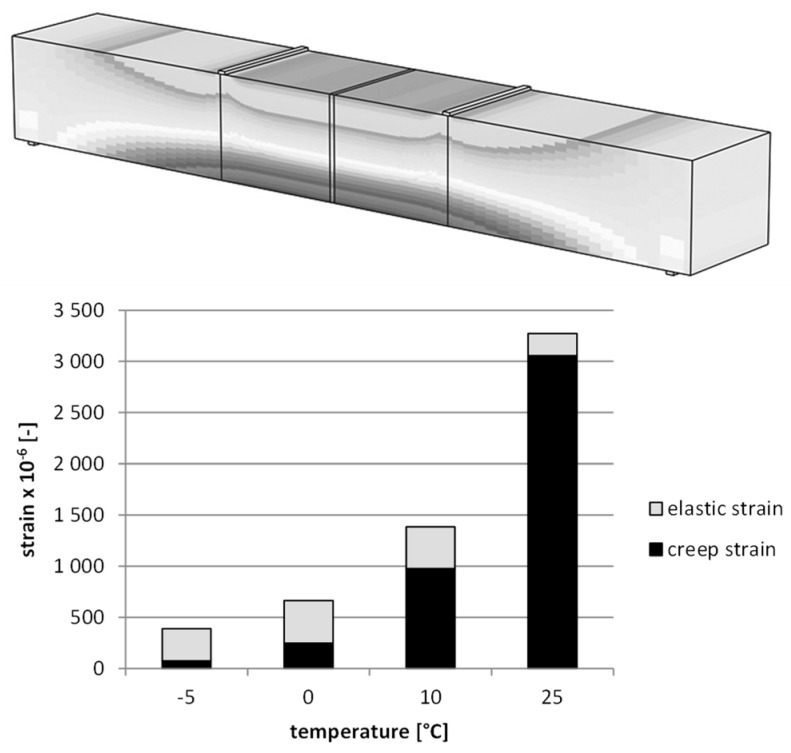
The deformation results for the static creep test (asphalt mixtures with an asphalt content of 4.5%).

**Figure 17 materials-12-02084-f017:**
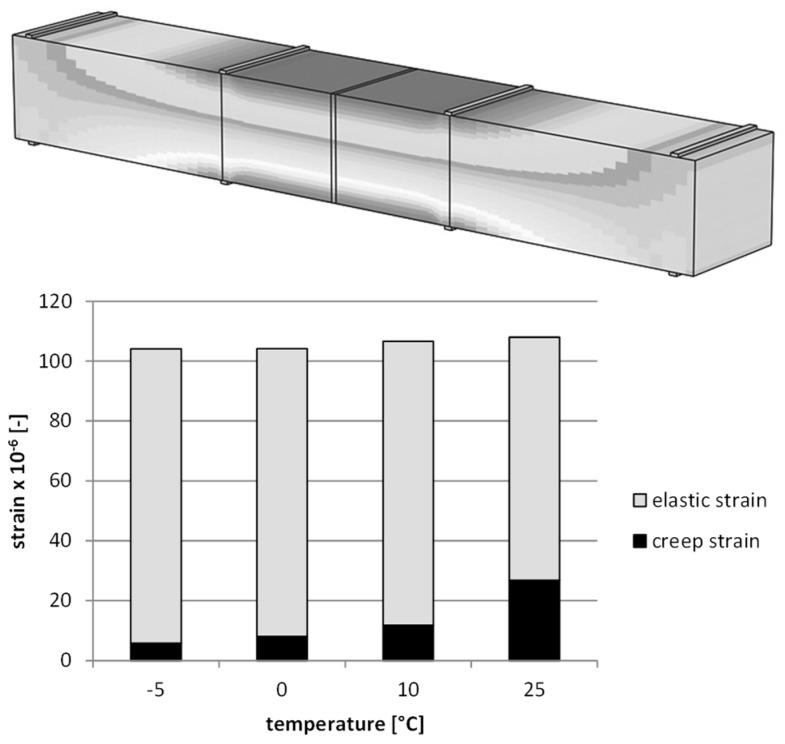
The deformation results for the fatigue dynamic test (asphalt mixtures with an asphalt content of 4.5%).

**Table 1 materials-12-02084-t001:** Rheological parameters of the Burger’s model obtained for the creep 4BP study.

Temperature	−5 °C	0 °C	10 °C	25 °C
asphalt content: 4.0%				
*E*_1_ (MPa)	5470	3619	1498	607
*η*_1_ (MPa·s)	1,497,854	769,973	308,374	121,832
*E*_2_ (MPa)	4311	3250	2235	1213
*η*_2_ (MPa·s)	1,881,548	821,849	34,011	28,569
asphalt content: 4.5%				
*E*_1_ (MPa)	5733	3902	1762	742
*η*_1_ (MPa·s)	1,570,013	830,165	362,676	148,891
*E*_2_ (MPa)	4519	3504	2628	1483
*η*_2_ (MPa·s)	1,972,192	886,096	40,000	34,914
asphalt content: 5.3%				
*E*_1_ (MPa)	5135	3474	1601	485
*η*_1_ (MPa·s)	1,406,134	739,012	329,545	97,405
*E*_2_ (MPa)	4047	3119	2388	970
*η*_2_ (MPa·s)	1,766,333	788,801	36,346	22,841

**Table 2 materials-12-02084-t002:** Rheological parameters of the Burger’s model obtained for the dynamic 4BP study.

Temperature	−5 °C	0 °C	10 °C	25 °C
asphalt content: 4.0%				
*E*_1_ (MPa)	26,494	20,164	11,703	8001
*η*_1_ (MPa·s)	23,454	17,520	9441	4327
*E*_2_ (MPa)	31,980	23,912	12,865	5797
*η*_2_ (MPa·s)	5165	3399	870	132
asphalt content: 4.5%				
*E*_1_ (MPa)	27,770	21,740	13,764	9778
*η*_1_ (MPa·s)	24,584	18,890	11,103	5288
*E*_2_ (MPa)	33,521	25,781	15,130	7084
*η*_2_ (MPa·s)	5414	3665	1023	161
asphalt content: 5.3%				
*E*_1_ (MPa)	24,871	19,353	12,507	6397
*η*_1_ (MPa·s)	22,018	16,816	10,089	3459
*E*_2_ (MPa)	30,022	22,950	13,748	4634
*η*_2_ (MPa·s)	4849	3263	930	105

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
