# Peer review of "Viscoelastic Parameters of Asphalt Mixtures Identified in Static and Dynamic Tests"

_materials, 2019, doi:10.3390/ma12132084_

Round 1
Reviewer 1 Report
The submitted manuscript analyzed the behaviors of the asphalt mixtures under the static and dynamic loading for a four-point bending beam. The topic is interesting, and the methodologies used in the study is also good to prove the proposed objectives in this manuscript. However, there are some technical comments could be considered, in order to ensure the manuscript could more readable for the potential readers of the journal.
(1) The abstract of the manuscript should be re-written, at least deep revise. Typically, the abstract is a summary of the article. The main objectives, the applied methodologies, and the brief conclusions should be included, but I feel the organization of the abstract is confusing...
(2) If possible, please try to improve the introduction section of this manuscript. Asphalt and asphalt mixture is typically viscoelastic materials in civil engineering (flexible pavement). There are some studies analyzed the mechanical behaviors of the flexible pavement under various loading conditions by using analytical, experimental approaches, for example, (i) Applied Mathematical Modelling, 61, pp.726-743. (ii) Road Materials and Pavement Design, 9(1), pp.31-57. (iii) Frontiers of Structural and Civil Engineering, 13(1), pp.110-122. (iv) Journal of Engineering Mechanics, 135(6), pp.517-528. In my opinion, for your paper, it is a good study to reveal the effects of viscoelastic parameters of asphalt mixtures via lab experiments and simulations, thus, it is necessary to enhance the background description from these two aspects.
(3) Revise the format of Equation (1): please remove "--> minimum" from the equation. You only need to explain it in the text. The expression is very strange!
(4) The regression formula is used in Figure 13, please clarify the related regression formula and the regression coefficient. Otherwise, it is a bit of confusing.
(5) Please provide the related citations for you cited equations, for example, Equation (17) is a basic function for the dynamic analysis in FEM, but I did not saw the related citation.
(6) The details of methodologies for the FEM simulation should be demonstrated, for example, the mesh technologies, the element properties you employed, etc.. I think this information is essential for the FEM modeling, although you have applied very scientific analyses.
Author Response
Reviewer I
Reviewer's comment:
(1) The abstract of the manuscript should be re-written, at least deep revise. Typically, the abstract is a summary of the article. The main objectives, the applied methodologies, and the brief conclusions should be included, but I feel the organization of the abstract is confusing...
Author’s response:
We corrected the abstract accordingly.
Reviewer's comment:
(2) If possible, please try to improve the introduction section of this manuscript. Asphalt and asphalt mixture is typically viscoelastic materials in civil engineering (flexible pavement). There are some studies analyzed the mechanical behaviors of the flexible pavement under various loading conditions by using analytical, experimental approaches, for example, (i) Applied Mathematical Modelling, 61, pp.726-743. (ii) Road Materials and Pavement Design, 9(1), pp.31-57. (iii) Frontiers of Structural and Civil Engineering, 13(1), pp.110-122. (iv) Journal of Engineering Mechanics, 135(6), pp.517-528. In my opinion, for your paper, it is a good study to reveal the effects of viscoelastic parameters of asphalt mixtures via lab experiments and simulations, thus, it is necessary to enhance the background description from these two aspects.
Author’s response:
We included in the Introduction section the proposed references.
Reviewer's comment:
(3) Revise the format of Equation (1): please remove "--> minimum" from the equation. You only need to explain it in the text. The expression is very strange!
Author’s response:
We removed it.
Reviewer's comment:
(4) The regression formula is used in Figure 13, please clarify the related regression formula and the regression coefficient. Otherwise, it is a bit of confusing.
Author’s response:
In Fig. 13, there is no regression function, while in Fig. 15, linear regression functions have been used that describe well the relationship between viscosity parameters and the damping coefficient. High correlation coefficients (close to 1) were obtained for this function. The article has been supplemented with this information.
Reviewer's comment:
(5) Please provide the related citations for you cited equations, for example, Equation (17) is a basic function for the dynamic analysis in FEM, but I did not saw the related citation.
Author’s response:
We included the appropriate citation for these equations.
Reviewer's comment:
(6) The details of methodologies for the FEM simulation should be demonstrated, for example, the mesh technologies, the element properties you employed, etc.. I think this information is essential for the FEM modeling, although you have applied very scientific analyses.
Author’s response:
We described in more details the FEM modeling.
In addition, the authors have adjusted some of the tasks to improve the grammar and form of the English language.

Reviewer 2 Report
The manuscripts named “Viscoelastic parameters of asphalt mixtures identified 2 in static and dynamic tests” investigates the determination of viscoelastic parameters under different loading conditions and various asphalt contents through four-point bending test. The authors used a Burger model which consists of a spring, which accounts for elastic behavior, a Maxwell element, which is responsible for delayed elastic behavior, and a dashpot element which carries the viscous response of asphalt mixture. After calculating the Burger model parameters, numerical modeling was carried out to further study the effect of temperature and loading conditions. This reviewer found this research study very interesting and applying the comments below could make the manuscript ready for publication.
The idea of considering the response of asphalt mixture both at the loading and the unloading parts is brilliant. The authors need to back this up using the appropriate literature review. For instance:
1) Jahangiri, Behnam, Mohammad M. Karimi, and Nader Tabatabaee. "Relaxation of hardening in asphalt concrete under cyclic compression loading." Journal of Materials in Civil Engineering 29, no. 5 (2016): 04016288.
2) Karimi, Mohammad M., Nader Tabatabaee, Behnam Jahangiri, and Masoud K. Darabi. "Constitutive modeling of hardening-relaxation response of asphalt concrete in cyclic compressive loading." Construction and Building Materials 137 (2017): 169-184.
3) Karimi, Mohammad M., Nader Tabatabaee, H. Jahanbakhsh, and Behnam Jahangiri. "Development of a stress-mode sensitive viscoelastic constitutive relationship for asphalt concrete: experimental and numerical modeling." Mechanics of Time-Dependent Materials 21, no. 3 (2017): 383-417.
Please describe the software which is used to perform numerical (FE) analysis.
Some editorial comments:
“The procedure of identification of viscoelastic material of asphalt mixtures used in pavements features depending on temperature and load conditions was described”. Rewrite.
“on temperature was found for the two test”. Rewrite (Tests)
Page 3-Line 106: “It consist in…” Rewrite.
Authors should pay attention to the paragraph indentation. There are many of indents which are not really needed.
Figure 13: Different markers could be used to distinguish the asphalt contents.
This reviewer would suggest using the bullet points to organize and summarize the conclusion and findings.
Author Response
Reviewer II
Reviewer's comment:
The idea of considering the response of asphalt mixture both at the loading and the unloading parts is brilliant. The authors need to back this up using the appropriate literature review. For instance:
1) Jahangiri, Behnam, Mohammad M. Karimi, and Nader Tabatabaee. "Relaxation of hardening in asphalt concrete under cyclic compression loading." Journal of Materials in Civil Engineering 29, no. 5 (2016): 04016288.
2) Karimi, Mohammad M., Nader Tabatabaee, Behnam Jahangiri, and Masoud K. Darabi. "Constitutive modeling of hardening-relaxation response of asphalt concrete in cyclic compressive loading." Construction and Building Materials 137 (2017): 169-184.
3) Karimi, Mohammad M., Nader Tabatabaee, H. Jahanbakhsh, and Behnam Jahangiri. "Development of a stress-mode sensitive viscoelastic constitutive relationship for asphalt concrete: experimental and numerical modeling." Mechanics of Time-Dependent Materials 21, no. 3 (2017): 383-417.
Author’s response:
We included these publication in the manuscript.
Reviewer's comment:
Please describe the software which is used to perform numerical (FE) analysis.
Author’s response:
We used SOLIDWORKS-COSMOS/M software. We mentioned about it in the manuscript.
Reviewer's comment:
Some editorial comments:
“The procedure of identification of viscoelastic material of asphalt mixtures used in pavements features depending on temperature and load conditions was described”. Rewrite.
“on temperature was found for the two test”. Rewrite (Tests)
Author’s response:
We corrected the abstract accordingly.
Reviewer's comment:
Page 3-Line 106: “It consist in…” Rewrite.
Author’s response:
We removed this odd sentence.
Reviewer's comment:
Authors should pay attention to the paragraph indentation. There are many of indents which are not really needed.
Author’s response:
We removed the unnecessary paragraph indentation.
Reviewer's comment:
Figure 13: Different markers could be used to distinguish the asphalt contents.
Author’s response:
Figure 13 has been corrected taking into account the relevant markers.
Reviewer's comment:
This reviewer would suggest using the bullet points to organize and summarize the conclusion and findings.
Author’s response:
Conclusions were presented in more concise form using the bullet points.
In addition, the authors have adjusted some of the tasks to improve the grammar and form of the English language.
